# Optimised generation of iPSC-derived macrophages and dendritic cells that are functionally and transcriptionally similar to their primary counterparts

Susan Monkley[1]*, Jayendra Kumar Krishnaswamy[2¤a], Melker Göransson[2], Maryam Clausen[3], Johan Meuller[3¤b], Kristofer Thörn[1], Ryan Hicks[4], Stephen Delaney[2], Louise Stjernborg[3]

1 Translational Science and Experimental Medicine, Research and Early Development, Respiratory and Immunology (R&I), BioPharmaceuticals R&D, AstraZeneca, Gothenburg, Sweden, 2 Bioscience, Research and Early Development, Respiratory and Immunology (R&I), BioPharmaceuticals R&D, AstraZeneca, Gothenburg, Sweden, 3 Discovery Biology, Discovery Sciences, BioPharmaceuticals R&D, AstraZeneca, Gothenburg, Sweden, 4 BioPharmaceuticals R&D Cell Therapy, Research and Early Development, Cardiovascular, Renal and Metabolism (CVRM), BioPharmaceuticals R&D, AstraZeneca, Gothenburg, Sweden

¤a Current address: Galderma Pharma SA, Lausanne, Switzerland
¤b Current address: Mölnlycke Healthcare, Gothenburg, Sweden
* susan.monkley1@astrazeneca.com

## Abstract

Induced pluripotent stem cells (iPSC) offer the possibility to generate diverse disease-relevant cell types, from any genetic background with the use of cellular reprogramming and directed differentiation. This provides a powerful platform for disease modeling, drug screening and cell therapeutics. The critical question is how the differentiated iPSC-derived cells translate to their primary counterparts. Our refinement of a published differentiation protocol produces a CD14+ monocytic lineage at a higher yield, in a smaller format and at a lower cost. These iPSC-derived monocytes can be further differentiated into macrophages or dendritic cells (DC), both with similar morphological and functional profiles as compared to their primary counterparts. Transcriptomic analysis of iPSC-derived cells at different stages of differentiation as well as comparison to their blood-derived counterparts demonstrates a complete switch of iPSCs to cells expressing a monocyte, macrophage or DC specific gene profile. iPSC-derived macrophages respond to LPS treatment by inducing expression of classic macrophage pro-inflammatory response markers. Interestingly, though iPSC-derived DC show similarities to monocyte derived DC, they are more similar transcriptionally to a newly described subpopulation of AXL$^+$ DC. Thus, our study provides a detailed and accurate profile of iPSC-derived monocytic lineage cells.

**Data Availability Statement:** RNAseq data for all samples is available from ArrayExpress with the accession number E-MTAB-9670.

**Funding:** The author(s) received no specific funding for this work. AstraZeneca provided support in the form of salaries for authors [SM, JKK, MG, MC, JM, KT, RH, SD, LS ], as well as research materials. Employees of AstraZeneca, as authors, participated in the study design, data collection and analysis, decision to publish, or preparation of the manuscript. AstraZeneca reviewed the publication, without influencing the opinions of the authors, to ensure medical and scientific accuracy, and the protection of intellectual property. All authors had access to all data in the study, and approved the decision to submit the manuscript for publication. The specific roles of the authors are articulated in the 'author contributions' section.

**Competing interests:** AstraZeneca is a global biopharmaceutical company specialising in the discovery, development, manufacturing and marketing of prescription medicines. AstraZeneca is a global biopharmaceutical company specialising in the discovery, development, manufacturing and marketing of prescription medicines. At the time of the study all authors were employees of AstraZeneca. This does not alter our adherence to PLOS ONE policies on sharing data and materials. While 2 authors are now employees of Mölnlycke Healthcare and Galderma Pharma SA, neither of these companies provided any funding or input towards this study.

## Introduction

Monocytes, macrophages and dendritic cells are part of the mononuclear phagocyte system of innate immunity with monocytes being the precursors to distinct sub-populations of macrophages and DC. They are found in blood as well as throughout the body as resident populations in many organs including the brain, skin, liver, lung, kidney, and heart. They are crucial for both the control of pathogens and initiation of immune responses and the support of tissue functions. As well as being targets for immunotherapy [1], they have been implicated as key players in both immune related diseases (autoimmune, inflammatory and infection) and non-immune diseases such as neurodegeneration, cardiovascular disease and cancer [2–4]. As such there has been a concerted effort over many years to study the development and function of these cells, but this has been hampered by difficulty in obtaining sufficient numbers of tissue resident macrophages and DC [5]. As a result, most experiments have been performed using transformed cell lines (THP1, RAW264.7), mouse-derived cells (bone marrow-derived macrophages) or use of differentiated blood monocytes (monocyte-derived macrophages and dendritic cells). However, there are drawbacks to the use of these alternatives. Cell lines are prone to contamination by other lines, genetic drift, are often karyotypically abnormal [6] and exhibit functional limitations [7]. Mouse bone marrow derived DC and macrophages have a number of functional differences to conventional murine DC as well as human DC and macrophages [8, 9]. Furthermore, monocyte derived macrophages and DC may not be reflective of their tissue derived equivalents [10–13] and do not allow for simple genetic manipulation.

The ability to use human pluripotent stem cells to generate a wide range of cell types has provided a potential alternative source of monocytes, macrophages and dendritic cells of human origin [14]. This method relies on differentiation of iPSC to a monocyte population which can be further differentiated into either macrophages or dendritic cells. The advantage provided by iPSC-derived cells is that they are of human origin, can be genetically manipulated with ease prior to differentiation or can be from patient derived cells recapitulating their genetic status and thus can be generated repeatedly, reproducibly and in relatively large quantities. This provides a powerful platform for disease modeling, drug screening and cell therapeutics. The critical question is whether the differentiated iPSC-derived cells translate to their primary counterparts.

Based on the published method [14] we have refined the differentiation conditions to improve yields, efficiency and cost. The differentiation protocol produces a monocytic lineage cells (CD14$^+$ cells) that are further differentiated into macrophages or dendritic cells. We demonstrate that the iPSC-derived macrophages (iPSdM) and iPSC-derived DC (iPSdDC) have comparable functional profiles to their primary counterparts. iPSdM can phagocytose bacteria and respond to bacterial lipopolysaccharides (LPS) similarly to blood monocyte derived macrophages (MDM). iPSdDC express key DC surface markers, acquire and process antigens, mature in response to LPS and can efficiently activate T cells.

Additional confirmation of the nature of these cells was obtained by performing RNA sequencing (RNAseq) to provide insight into both the differentiation process and to compare these cells to their commonly used alternatives (monocyte derived macrophages and dendritic cells). This is the first study to report the transcriptional profiling of iPSC-derived monocytes and DC. Analysis of this data demonstrates a complete switch from iPSC to cells expressing monocytic specific gene profiles. RNAseq analysis of iPSdM, shows that these cells respond to LPS treatment by inducing expression of classic macrophage pro-inflammatory response genes. These data support iPSdM as a suitable macrophage model and shows the value of transcriptional profiling for other cell models. We also demonstrate that while iPSdDC show similar transcriptional profiles to their monocyte derived counterparts, they more closely resemble a newly

identified human DC population, the AXL+ DC [15], raising the possibility that these cells may be a valuable source of cells for further investigating the biology of this new DC population.

## Materials and methods

### Monocytic lineage cell differentiation method

The monocytic lineage differentiation protocol is based on the 5 sequential step protocol published by Yanagimachi et al 2013 [14], by which mature macrophages and dendritic cells are differentiated from human induced pluripotent stem cells (iPSC) via the monocyte stage. The iPSC line used (r-iPDSC-1j) which is derived from human foreskin fibroblast from ATCC using mRNA programming has been described previously [16]. To generate a more robust protocol with a higher yield of CD14+ cells, the protocol was modified including using embryoid bodies (EBs) of known cell number ($30x10^3$) instead of randomly sized cell clusters and growing the EBs in 6-well plates instead of 100 mm Petri dishes.

At day 0, the differentiation and generation of EBs was started simultaneously by seeding undifferentiated single cell iPS cells (at a concentration of 30K cells per well in a 96 Ultralow attachment plate with v-shaped wells (#3896, Corning) in 100 μl media (mTeSR1 complete with 10 μM Y-27632 (ROCK inhibitor, #181130, Calbiochem,) and 80 ng/ml BMP-4 (#314-BP, R&D Systems). The plate was centrifuged at 200xg, for 3 min. at room temperature. On day 2 the EBs were transferred to growth factor-reduced Matrigel coated wells (#356230, Corning), five EBs per well in 6 well plates (#3506, Corning) in the same media as described above.

For the rest of the differentiation, the Yanagimachi protocol was followed. Twice weekly, from day 18 to day 40, the suspension cells were positively sorted by autoMACS using CD14 + MicroBeads (Miltenyi Biotech) generating a total of 6 to 7 batches of CD14+ monocytic lineage-directed cells (Fig 1A) providing a continuous supply of cells that can be used directly or cryopreserved for later use. All batches of these CD14+ monocytes (iPSdMo) were further differentiated into either macrophages (iPSdM) or dendritic cell (iPSdDC) as described below. A total of 6 differentiation runs were performed, three for each final cell type (macrophage/dendritic cell). Cells originating from harvests 2, 4 and 6 (approx d21, d28 and d35) were used for

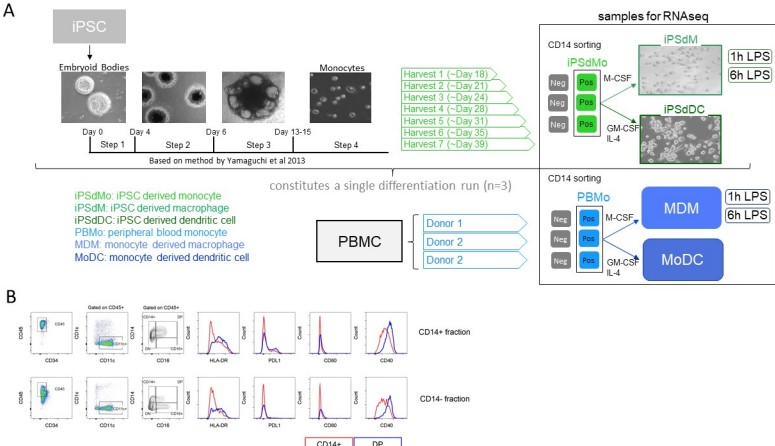

**Fig 1. Generation of iPSC-derived monocyte, macrophages and dendritic cells and their comparison to blood derived counterparts.** (A) Schematic illustrating the protocol and the samples generated including and used for downstream analysis. Also shown are representative images of the iPSC-derived cells both during the differentiation process and at terminal differentiation stages. (B) FACS analysis of the 2 fractions from the CD14 sorting step. Top row; CD14+ and bottom row; CD14- fractions respectively. Antibodies used are shown on the plots. Blue; CD16+ (DP; double positive) and red; CD16- fractions (CD14+).

RNA sequencing. Cells derived from the other harvest times were used for FACS analysis and functional assays (Fig 1A).

## In vitro differentiation of human monocyte derived macrophages, iPSC-derived macrophages, human monocyte derived dendritic cells and iPSC-derived dendritic cells

Venous blood from healthy human volunteers (ethical approval number from the local AZ Ethics committee in Gothenburg, Sweden, Dnr T705-14 Ad 033–10), collected in EDTA tubes, was diluted 1:2 in PBS, layered on top of Lymphoprep solution (Fresenius Kabi) and fractionated by centrifugation to recover the peripheral blood mononuclear cell (PBMC) fraction. CD14$^+$ cells were isolated from the human blood PBMC fraction using the MACS huCD14 microbead kit (Miltenyi biotech), according to the manufacturer's protocol. These CD14 + peripheral blood monocytes are hereafter referred to as PBMo (Fig 1A)

For macrophage differentiation, CD14$^+$ cells (either blood or iPSC-derived) were resuspended in RPMI1640 + 10%FCS containing 100ng/ml human M-CSF (R&D Systems), seeded in plates and differentiated for 7 days giving rise to MDM or iPSdM (Fig 1A).

For DC differentiation, CD14$^+$ cells (either blood or iPSC-derived) were resuspended in cell medium (Complete StemPro-34) complemented with GM-CSF to 80 ng/ml and IL-4 to 40 ng/ml, seeded in ultra-low attachment plates (at 1.2 x 10$^5$ cells/cm$^2$) and allowed to further differentiate into dendritic cells for 6–9 days giving rise to MoDC or iPSdDC (Fig 1A). Maturation of the dendritic cells were achieved by supplementing the media with 100 ng/ml LPS and 0.2 ng/ml TNFα.

For the RNA seq experiments, 2.5 x 10$^5$ CD14$^+$ cells were seeded in 500μl media in 48 well plates and differentiated into either macrophages or DC.

## Macrophage stimulation and mediator analysis

Stimulation experiments were performed on macrophages using 100ng/ml LPS from E. coli (serotype O127:B8, Sigma) for 1h and 6h respectively. Plates were spun at 300g for 5 min and media was harvested and stored at -80˚C until analyzed using the human V-plex proinflammatory panel I (Meso Scale Diagnostics). Statistical testing was done using unpaired 2-tailed T-test.

## pHrodo *S. aureus* phagocytosis assay

Human MDM and iPSdM were differentiated, as described above, in optical 96 well plates (μClear, Greiner) at a density of 100 000 cells/well and their capacity to phagocytose bacteria was measured using *S. aureus* conjugated to pHrodo (Invitrogen). Extracellular or surface bound pHrodo$^+$ bacteria are non-fluorescent but turn bright red when phagocytosed. pHrodo particles were resuspended in RPMI1640 with 25mM HEPES at a concentration of 1mg/ml and sonicated thoroughly. After removal of cell media, 100μl resuspended particles were added to each 96 well. Control wells contained cells only, beads only or beads with cells pretreated with 10 μM cytochalasin D (Sigma-Aldrich). Cells were incubated for 2h at 37˚C and then fixed using 2% phosphate-buffered formaldehyde for 5 min. After a PBS wash, nuclei were counterstained by Hoechst stain 1:10000 (Invitrogen). Phagocytosis was measured by ImageXpress (Molecular Devices) at 560/585nm. Experiments were performed in triplicates in 3 monocyte donors and in 3 separate IPSC differentiation experiments. Data were calculated as % positive cells based on total numbers of nuclei. Statistical testing was done using one way ANOVA with Sidak's multiple comparisons test.

## Mixed leucocyte reaction

Naïve CD4[+]/CD8[+] T cells were isolated from fresh human PBMC using the Pan T Cell Isolation Kit II (Miltenyi # 130-091-156) and subsequently labelled with CFSE (Vybrant® CFDA SE Cell Tracer Kit, Invitrogen, cat# V12883) according to manufacturer's instructions. Dendritic cells were differentiated from blood or iPSC-derived monocytes as described above. Immature or LPS matured DC were mixed with CFSE labelled naïve T-cells in a 1:2.5 ratio and incubated for 6 days at 37˚C, 5% CO2 to stimulate T-cell proliferation. At the end of the incubation time, samples were analyzed using FACS to determine the number of CFSE positive cells in the CD3/CD8 population. Statistical testing was done using paired 2-tailed T-test.

## Antigen uptake and processing kinetics

Immature or LPS matured DC were incubated with DQ[TM] Ovalbumin (ThermoFisher Scientific; 10ug/mL) at either 4˚C or 37˚C. At indicated time points, cells were collected and analyzed using FACS. Statistical testing was done using 2-way Anova with Sidak's multiple comparisons test.

## FACS analysis

Cells were collected and washed once in PBS before staining with LIVE/DEAD Fixable Aqua (ThermoFisher Scientific) stain for 15 min at RT. Following this, cells were washed once with PBS containing EDTA (2 mM) and BSA (0.1%) and incubated with fluorochrome-conjugated antibody cocktail for 30 min at 4˚C. Cells were subsequently washed once and resuspended in FACS buffer before analysis on the BD Fortessa flow cytometer. Anti-human CD14 (M5E2), CD16 (3G8), CD40 (5C3), CD80 (L307.4), PDL1/CD274 (MIH1), HLA-DR (G46-6), CD11c (B-ly6), CD1c (F10/21A3), CD45 (2D1), CD34 (581) monoclonal antibodies were purchased from BD Biosciences.

## RNA isolation, library, sequencing

Cells for sequencing were lysed in 350µl RLT plus buffer and RNA was prepared using the RNeasy mini plus kit (Qiagen) according to the manufacturer's instructions. The quality of the RNA was assessed by a Fragment Analyzer (Advanced Analytical Technologies). Samples with RNA integrity number >8.8 were used for library preparation. One microgram of total RNA was used for each library. RNA samples were processed with Illumina TruSeq Stranded mRNA Library prep kit selecting for poly(A) tailed RNA following the manufacturer's recommendations. All libraries were quantified with the Fragment Analyzer (Advanced Analytical Technologies) and Qubit Fluorometer (Invitrogen) and adjusted to the appropriate concentration for sequencing. Indexed libraries were pooled and sequenced at a final concentration of 1.6 pM on an Illumina NextSeq 500 high-output run using paired-end chemistry with 75 bp read length.

## Bioinformatics analysis

Sequencing was done in 2 batches using the same methodology and equipment for each. RNA sequencing fastq files were processed separately using bcbio-nextgen (version 0.9.7) where reads were mapped to the human genome build hg38 (GRCh38.79) using hisat2 (v 2.0.5) [17] to generate bam files which were then used by featureCounts (v1.4.4) to generate counts of reads for each gene. TPM (transcripts per million) data tables for genes and transcripts were generated using Sailfish (v 0.10.1) mapping and counting package [18]. The final count and TPM matrices were generated by joining sample columns for each sequencing batch by gene/

transcript. Batch correction using sva [19] was tested but did not improve the outcome of differential gene expression and so was not used.

Differential gene expression was determined using the DESeq2 package [20] for comparisons between the iPSC versus iPSdMo (CD14+), iPSdMo (CD14-) versus iPSdMo (CD14+), as well as between the three combinations of iPSC versus blood derived cells. As the numbers of differentially expressed genes (DEG) was very high for all comparisons presented here a multiple testing adjusted p-value (FDR; false discovery rate) cutoff of 0.01 was routinely used. Additional cutoffs of genes with TPM >2 or log2 fold change (LFC) cutoffs are referred to in the text where applicable.

Visualisations (PCA plots, bar charts) were implemented in R using standard packages (ggplot2) or ArrayStudio (v10.0, QIAGEN Inc., Omicsoft). PCA plots and correlation scatter plots were generated with rlog transformed normalized counts generated by DESeq2.

Functional enrichment was performed using IPA (QIAGEN Inc., https://www.qiagenbio informatics.com/products/ingenuitypathway-analysis) using the Core Analysis tool. The Disease and Functions results with p-value <0.05 were extracted for further filtering. The activation status of the functions/pathways were predicted using IPA Upstream Regulator Analysis Tool by calculating a regulation Z-score and an overlap p-value, which were based on the number of known target genes of interest pathway/function, expression changes of these target genes and their agreement with literature findings. The detailed descriptions of IPA analysis are available under "Downstream Effects Analysis "on the IPA website (https://www.qiagen bioinformatics.com/products/ingenuitypathway-analysis). Redundant functions were removed, generating a long list of enriched functions (S4 Table in S1 File). Of these 15 of the most informative and relevant functions were used to generate plots for Figs 4–6).

The RNA sequencing data has been deposited in ArrayExpress with the accession number E-MTAB-9670.

## Comparison to published data

The TPM (transcripts per million) expression data was downloaded for GSE94820 [15] from GEO and the 219 genes that were reported as discriminating the different DC populations (S2 Table in S1 File, ref 15] with AUC value > 0.85 were used for next steps. The TPM expression data for the same genes were also extracted for the DC populations from this study (MoDC and iPSdDC). As the 2 studies were performed and processed using different methods it is not possible to compare them directly so the value in each study were convert to Z-scale per gene and then combined. A heatmap with hierarchical clustering (using complete linkage, Pearson correlation and robust centre scale normalisation) was generated for all the samples in the combined dataset using ArrayStudio v10.0 (QIAGEN Inc., Omicsoft).

## Results

### Induced pluripotent stem cell derived macrophages and dendritic cells– 5-step protocol

Our improved protocol to generate iPSC-derived CD14$^+$ cells was based on a published protocol [14]. One improvement was the change from randomly sized colonies to optimized reproducible sized embryoid bodies generated from single cells. This enabled simultaneous EB formation and differentiation at day 0, which gave a robust and highly efficient protocol with reproducible sized EB and a higher CD14+ cell yield. We also evaluated different culture formats and found that the same cell yield (~8 x10$^6$ cells per harvest) could be obtained from a single 6-well plate (58cm$^2$, 119 ml media) with our improved protocol as was obtained with 4x

100 mm culture dishes (220cm$^2$, 336 ml media) using the original protocol. This is an increase in yield of 4.3x per cm$^2$ culture surface and has significant cost savings (1000USD per run) of expensive growth factors and cytokines. The new protocol produces 2.5–4 x10$^6$ CD14+ monocytes per harvest and up to 3.5 x10$^7$ cells over the course of a single differentiation run which compares very well to other recently published methods [21, 22].

The differentiation run was performed three times to capture variability between differentiations and for each differentiation run 6–7 harvests of the suspension monocytes were collected (Fig 1A). This allowed transcriptomic comparisons between runs and harvests to assess the reproducibility of the method. In general harvests 2, 4 and 6 were used for RNAseq while others were used for functional assays.

## iPSC-derived monocytes are phenotypically distinct from their stem cell precursors

To further understand the nature of the monocyte-like suspension cells released after the iPSC differentiation method, both the CD14+ and CD14- iPSdMo fractions were characterized using flow cytometry. A representative example in Fig 1B shows both the CD14+ and CD14- cells were CD45+ CD34- indicating that these cells are of hematopoietic origin but are no longer stem cell progenitors [23, 24]. CD14+ iPSdMo were 95% pure and consisted primarily of CD11c + CD1c- cells which could further be segregated into CD16+ (DP) and CD16- fractions (CD14 +), similar to the subsets of primary human blood monocytes [25]. The CD14- fraction also consisted of a CD16+ fraction reminiscent of CD14- CD16+ non-classical human monocytes [26] but also contains other cells of unknown status (Fig 1B). CD14+ CD16- iPSdMo have an immature phenotype expressing low levels of HLA-DR and other co-stimulatory molecules like PDL1, CD80 and CD40 (Fig 1B), in line with previous studies on primary human monocytes [26].

## iPSC-derived macrophages phenotypically resemble blood monocyte derived macrophages

To test phagocytic function and inflammatory responses in our iPSdM and compare them to MDM, both were subjected to a pHrodo phagocytosis assay and stimulation with LPS. In the phagocytosis assay, MDM and iPSdM both showed extensive *S. aureus* internalization Fig confirming that while the iPSdM described are indeed able to phagocytose bacteria in a manner similar to blood monocyte derived macrophages (MDM) it is to a significantly lesser extent (Fig 2A). Additionally, both MDM and iPSdM responded readily to LPS stimulation by secreting key inflammatory markers IL-6, and TNFα into the cell media, while IL-1β and IL-12p70 levels where more modest in both cell types (Fig 2B).

## Normal maturation, antigen uptake and antigen presentation processes seen in iPSC-derived DC

The iPSdDC are CD45+ CD11c+ CD1c+ as well as HLA-DR+. After LPS stimulation, iPSdDC undergo a maturation process and upregulate co-stimulatory molecules such as CD80 and CD40 while CD16, HLA-DR and PDL1 remain unaffected (Fig 2C). LPS matured iPSdDC were able to acquire and process antigen at slightly slower kinetics as compared to immature DC (Fig 2D). These differences were not due to increased apoptosis in LPS-activated iPSdDC. Activation of iPSdDC treated with LPS induced increased antigen presentation (Fig 2E) and inflammatory cytokine production (Fig 2F) indicating that these DC do undergo maturation and are capable of activating T cells in vitro in a manner similar to that previously reported for human monocyte-derived DC (MoDC) [27].

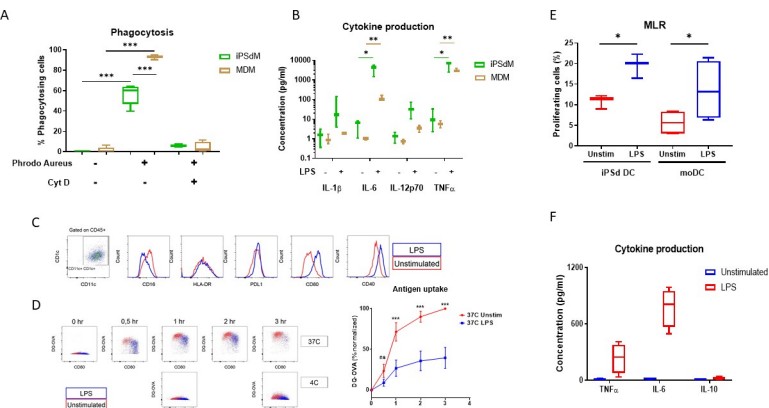

**Fig 2. Phenotypic characterization of macrophages and dendritic cells.** (A) Phagocytotic capacity measuring *S. aureus* internalization after 2h and (B) LPS-induced cytokine release both comparing MDM and iPSdM. (C) Gating strategy and surface marker expression and (D) antigen uptake and processing of iPSdDC before and after LPS maturation. (E) Comparison of the antigen presentation capability of both MoDC and iPSdDC measured by T-cell proliferation (Mixed leukocyte reaction; MLR) and (F) cytokine release from iPSdDC before and after LPS activation. *$p < 0.05$, **$p < 0.01$, ***$p < 0.001$.

## Transcriptomic analyses of iPSC-derived monocytes shows distinct transcriptional changes during the differentiation process

In line with recent studies, RNA sequencing was used to characterise the various iPSC-derived cell types and compare the transcriptomes of these to their blood derived counterparts [28, 29]. Reproducibility of the differentiation process was also investigated by including cells from three different harvests from three separate differentiation runs. These were sequenced along with the undifferentiated iPSCs and the blood derived cells from 3 different donors as comparators (Fig 1A). Differential gene expression analysis was used to compare the transcriptomes of the various cell types and the lists of differentially expressed genes can be found in S1–S5 Tables in S1 File.

Comparison of iPSC to the CD14+ iPSC-derived monocytes (iPSdMo) obtained after three or more weeks post-differentiation, identified over 6000 differentially expressed protein coding genes (greater than 4-fold difference in expression at FDR < 0.01; full list of DEG in S1 Table in S1 File) highlighting the substantial transcriptional changes the iPS cells undergo in order to become monocytes. This included the loss of expression of the pluripotency markers DNMT38, EPCAM and POU5F1 (OCT4) and the up-regulation of CD14 and FCGR3A/CD16 (Fig 3A).

Differential expression analysis comparing the CD14+ to CD14- iPSdMo populations identified 689 genes which were differentially expressed more than 4-fold (FDR < 0.01) (S2 Table in S1 File). This included a greater than 5-fold difference in expression levels of the CD14 transcript between these two groups, however the CD14- cells still expressed high levels of CD14 transcript (over 300 TPM on average) in keeping with a proportion of these cells possessing CD14 on the cell surface (Fig 1B, bottom row).

## Human iPSC-derived monocyte, macrophages and dendritic cells are transcriptionally similar to their blood derived counterparts

Transcriptional profiling was initially used to confirm that the differentiation protocol had achieved its desired aim in specifically differentiating iPSC into cells of a monocyte lineage including (Fig 3A). iPSdMo and iPSdM express the monocyte/macrophage markers CD14 and

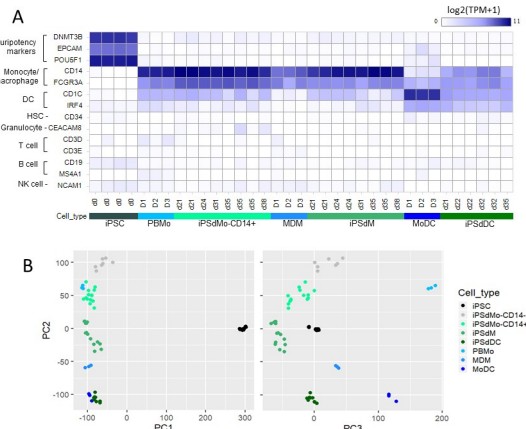

**Fig 3. Differentiation of iPSC into monocytes and derivatives results in loss of pluripotent markers and concomitant up regulation of lineage markers associated with differentiated cell types.** (A) Expression of lineage specific marker genes across all cell types., HSC; haematopoietic stem cell, NK cell; natural killer cell. Each column represents an individual sample and are grouped into cell type as shown. For the blood derived samples the sample has a donor designation (D1-3) for the IPSC-derived samples the sample has a designation that refers to the harvest day (Fig 1A). (B) Principal component analysis of all cells types generated in this study. PC1 v PC2 left panel, PC3 v PC2, right panel. Plots based on all genes.

FCGR3A(CD16) as do the PBMo and MDM (Fig 3A). iPSdDC and MoDC express CD1C and IRF4 as well as lower levels of CD14 and FCGR3A/CD16. None of the monocyte, macrophage or DC cells expressed any iPSC markers or any markers of cells from other hematopoietic lineages (Fig 3A).

Principal component analysis (PCA) was used to illustrate the sample relationships to one another based on their gene expression profiles (Fig 3B) and shows clearly that gene expression allows for grouping of the cells by cell type and origin. The first principal component (PC1) separates the pluripotent iPSC from all other (differentiated) cells while PC2 separates the different cell types (DC, Mo and Mac) and PC3 separates the iPSC-derived from blood derived cells (Fig 3B).

## iPSC-derived monocytes show distinct transcriptional signatures compared to blood derived monocytes but do not show major differences during differentiation to macrophages or DC

Differential gene expression profiling was also used to compare the transcriptomes of the iPSdMo, iPSdM and iPSdDC to their respective blood derived counterparts (S3–S5 Tables in S1 File respectively). Of the 13272 genes expressed (with TPM >2) in either iPSdMo or CD14 + PBMo, there was a positive correlation of gene expression levels between the 2 cells types (r = 0.97) with the majority (11308, 85.3%) expressed at similar levels (less than 4-fold difference at FDR < 0.01; Fig 4A). 1228 genes were expressed 4-fold higher in iPSdMo and 736 genes at 4-fold higher in PBMo.

Enrichment analysis of the DEG was used to determine which functional groups were associated with each cell type (S6 Table in S1 File and Fig 4B). Functions associated with iPSDMo relative to the PBMo (positive z-score) included processes related to cell cycle progression, cell migration and inflammatory response. Functions associated with PBMo relative to the iPSdMo (negative z-score) included activation/proliferation of leukocytes, apoptosis and antiviral response. Various functions associated with gene expression/protein translation were associated with both cell types perhaps suggesting cell type specific regulation of these processes.

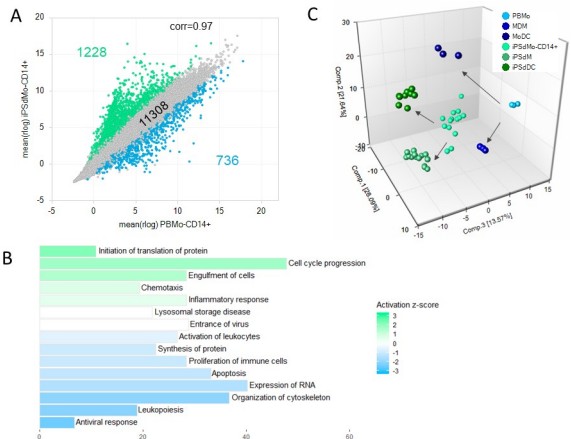

**Fig 4. iPSC-derived monocytes show many similarities to PBMC with respect to transcriptional profile and differentiation capacity.** (A) Correlation scatter plot of the average rlog expression values for PBMo versus iPSdMo. Only genes with TPM>2 in either group are shown. Genes with fold change >4 in green and genes with fold change < -4 in blue. (B) Summary of functional groups enriched for the 1964 DE genes. Bar length is the–log2 of the enrichment p-value and the colour represents the z-score with green indicating functions activated in iPSdMo relative to PBMo and blue bars those activated in PBMo relative to iPSdMo. (C) PCA plot of the various iPSdMo, iPSdM, iPSdDC, PBMo, MDM and MoDC samples. The arrows illustrate the progression of either PBMo or iPSdMo to either MDM/MoDC or iPSdM/iPSdDc respectively.

Despite any differences in gene expression between the blood derived and iPSC-derived monocytes, differentiation of either of these into macrophages by treatment with M-CSF or dendritic cells by GM-CSF/IL4 treatment resulted in very similar transcriptional changes as illustrated by the similar trajectories of these cells in principal component space (Fig 4C). It also resulted in a tighter clustering of the replicates.

## iPSdM are transcriptionally similar to MDM

Of the 11082 genes expressed (TPM > 2 in either iPSdM or MDM) there was a very strong correlation in expression levels of the majority of genes between the two cell types (r = 0.98), with 95% (10518 genes) expressed at similar levels (less than 4-fold difference at FDR < 0.01; Fig 5A). Only 564 genes were differentially expressed by at least 4-fold (303 up in iPSdM and 261 up in MDM).

Due to the relatively small number of differentially expressed genes identified between the two macrophage types, the functional enrichment analysis identified relatively few significantly enriched functional groups (S7 Table in S1 File). Those activated in the iPSdM relative to MDM were mostly functions associated with phagocytosis (including endocytosis and engulfment of cells; Fig 5B). The functional groups activated in MDM relative to iPSdM included inflammatory response as well as those associated with chemotaxis (cell movement, migration, and cytoskeleton organisation; Fig 5B).

One of the hallmarks of macrophages is their response to bacterial lipopolysaccharide (LPS) and the resulting expression of pro-inflammatory genes. LPS treatment of both MDM and iPSdM for 1 or 6 h resulted in the upregulation of specific pro-inflammatory genes in both cell types (Fig 5C). In addition, a PCA plot of the both iPSdM and MDM cells with and without LPS treatment demonstrates that both cell types also show very similar shifts in PC space upon treatment with LPS (Fig 5D).

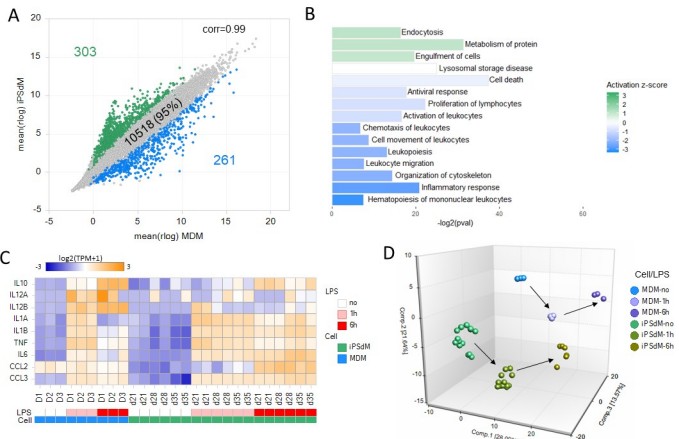

**Fig 5. iPSC-derived macrophages and MDM show similar transcriptional and activation profiles.** (A) Correlation scatter plot of the average rlog expression values for MDM versus iPSdM. Only genes with TPM>2 in either group are shown. Genes with fold change >4 in green and genes with fold change < -4 in blue. (B) Summary of functional groups enriched for the 1964 DE genes. Bar length is the -log2 of the enrichment p-value and the colour represents the z-score with green indicating functions activated in iPSdM relative to MDM and blue bars those activated in MDM relative to iPSdM. (C) Heatmap illustrating changes in expression of 10 LPS inducible genes (left) for both MDM and iPSdM following LPS stimulation. (D) PCA plot iPSdM and MDM samples with and without LPS stimulation. Grey arrows show the progression from no LPS to 1h LPS to 6h LPS.

## iPSC-derived DC while related to MoDC, are transcriptionally similar to Axl expressing DC

Comparing the expression of the 13023 genes (expressed at TPM >2 either for iPSdDC or MoDC) demonstrated there was a good correlation of gene expression levels (r = 0.98) with 89% (11603 genes) expressed at similar levels (less than 4-fold difference at FDR < 0.01). Of the differentially expressed genes, 928 genes were expressed 4-fold higher in iPSdDC and 497 at 4 fold higher in MoDC (Fig 6A). Enrichment analysis of the DEG (S8 Table in S1 File) identified 'activation of antigen presenting cells' in the iPSdDC relative to MoDC while 'activation of mononuclear leukocytes' was identified as being active in the MoDC relative to iPSdDC (Fig 6B). As for the iPSdM vs MDM comparison, 'inflammatory response' is activated in iPSdDC relative to the MoDC. Several functional groups that can be described as associated with migration and phagocytosis (migration of cells, phagocytosis, formation of cellular protrusions, organisation of cytoskeleton) have both positive and a negative z-scores suggesting that these 2 cell types may perform migration and phagocytosis via different mechanisms (Fig 6B).

In an effort to further characterise how the iPSdDC relate to other population of primary human DC, we utilised the single cell gene expression data generated by Villani and colleagues for groups of different blood DC subsets [15]. The study identified 6 groups of DC that are each defined by the expression of 5 genes. We compared the expression of these genes in our iPSdDC and iPSdMo to the MoDC and PBMo (Fig 6C). The MoDC were found to be most similar to the CD1C+ DC2 sub-population with some overlap with DC1. The PBMo and iPSdMo both show highest similarity to the DC3 sub-group. Most interestingly, we found that the iPSdDC we generated in this study are most transcriptionally similar to the DC5 group using this approach. When a longer list of 219 genes that were able to discriminate the 5 DC subtypes from each other [15] was used to cluster the DC cells from Villani and the DC cells from this study again the iPSdDC and DC5 cells cluster together (S1 Fig) alongside the DC6 cells this time as well, while the MoDC cluster with the DC1 in this time.

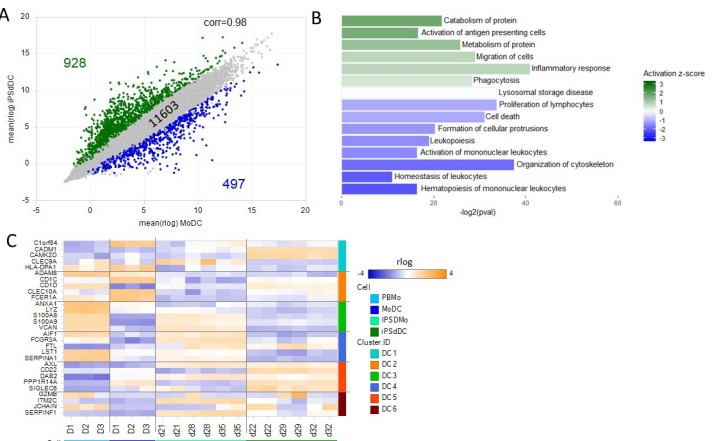

**Fig 6. iPSC-derived DC while similar transcriptionally to monocyte derived DC are more similar to blood derived DC.** (A) Correlation scatter plot of the average rlog expression values for MoDC versus iPSdDC. Only genes with TPM>2 in either group are shown. Genes with a fold change >4 are in green and genes with a fold change < -4 in blue. (B) Summary of functional groups enriched for the 3450 DE genes. Bar length is the -log2 of the enrichment p-value and the colour represents the z-score with green indicating functions activated in iPSdDC relative to MoDC and blue bars those activated in MoDC relative to iPSdDC. (C) Heatmap showing expression levels of genes identified in Villani as classifiers of the six DC subtypes across both types of monocytes and DC from this study.

## Transcriptional differences common to all iPSC-derived cells compared to blood derived counterparts

It became apparent when comparing the iPSC-derived cells to their blood derived counterparts that there were some differentially expressed genes that were common to all. Of the 3310 genes found to be differentially expressed between any of the iPSC-derived versus blood derived cells, about 10% (337) were common and in the same directions across all three pairs (S2 Fig). Of these, 259 genes were always significantly (LFC >2, FDR < 0.01) upregulated in the iPSC-derived cells as compared to the blood derived cells (S2A Fig). These genes are enriched for functions such as cell survival, cell-cell contact, migration of cells/cell movement, as well as activations of phagocytes and endocytosis. There is minimal overlap in the genes that are downregulated in iPSC-derived compared to blood derived cells (S2B Fig).

## Discussion

We have developed a substantially improved method for the differentiation of human iPSC to monocytic cells resulting in more reproducible, efficient and cheaper generation of large numbers of monocytes. Our method is based on improvements made to one of the earliest published methods for generation of iPSC-derived macrophages and dendritic cells and there have since been many other methods published for generating these cells [21, 22, 28, 29] using a variety of different approaches such as use of feeder cells for growth of iPSC [21], no pre-sorting of CD14 + cells prior to differentiation and use of hypoxic growth conditions [22]. However our method remains comparable in yield of CD14+ cells from a single harvest to the one more recent method where such comparable data was supplied [21], and in addition our method also allows for larger numbers of cells to be harvested from the culture over a period of weeks and although this has not been reported for other standard culture methods to date [21, 22, 29] there has been a recent report of large scale production of iPSC-derived macrophages [30].

Following the CD14 sorting, 2 populations of iPSdMo cells can be identified, CD14+ and CD14-. The CD14+ cells can efficiently be differentiated into both macrophages and dendritic

cells. The CD14- population are likely to contain a mix of undifferentiated, partially differentiated and inappropriately differentiated cells with unpredictable functional behaviours and as such are not suitable for direct characterisation. Further analysis of the other genes that differed between these 2 groups may suggest what other populations are found in the CD14- mix and possibly how the efficiency of CD14+ cell differentiation can be improved. Following further sorting and analysis they may however be of value for exploring functions of non-classical or intermediate human monocytes which are reported to play a role during inflammatory responses [26].

We have demonstrated that the CD14+ monocytes derived from iPSC can be efficiently differentiated into both macrophages and dendritic cells and that these cells are functionally and transcriptionally similar to primary monocyte-derived macrophage and dendritic cell types. The iPSdM show very similar LPS dependent inflammatory responses [31, 32], while the modest IL-1β response to LPS stimulation observed here in both MDM and iPSdM (Fig 2B) is consistent with a previous report showing that macrophages, in contrast to monocytes, need multiple stimuli to activate the inflammasome and potentiate IL-1β release [33]. In the phagocytosis assay the iPSdM exhibited a lower efficiency compared to MDM which is in line with previous reports comparing similar cell types [34]. It has been suggested that one of the reasons for the lower phagocytic activity in the iPSdM and related cell types is that they are skewed towards a more anti-inflammatory phenotype and perhaps more similar to tissue resident macrophages [34, 35]. This suggests they may therefore be potential models for not just MDM but the more difficult to obtain and disease relevant tissue-resident macrophages such as microglia, alveolar and kidney macrophages, Kupffer and Langerhans cells.

The iPSdDC display key DC cell surface marker expression [36] and are able to take up antigen although the uptake efficiency is reduced by LPS treatment which differs from that reported for murine DC [37]. We hypothesize that these differences are possibly due to differences in phagosome maturation in TLR activated iPSdDC. However, this does not significantly impair T cell stimulation capacity of IPSdDC. Antigen presentation of the iPSdDC was comparable if not better than MoDC following LPS stimulation.

In addition to these functional assays, RNA sequencing was also performed to allow a more thorough comparison between the different cell types. Such transcriptomic approaches have proven valuable in highlighting the similarities and differences between iPSC-derived and human sourced cells of many types [28, 29, 38] but this study is the first to report such an approach to compare iPSC-derived monocytes and DC to other sources of these cells.

Transcriptomic analysis of the different cells types was able to confirm both loss of key pluripotent markers from the differentiated cells as well as demonstrating the differentiation protocol specifically produced cells of the monocytic lineage with no contamination by other hematopoietic cell lineages (Fig 3A). It also allowed us to look at the reproducibility of generation of these cells from iPSC which was initially a key concern of this project. However, it was established that both for separate differentiation runs and for different harvest days the iPSC-derived monocytes were remarkably homogeneous and only became more so upon differentiation (Figs 3A, 3B and 4C).

Transcriptional analyses of all three iPSC-derived cells types (monocytes, macrophages and DC) as compared to their blood-derived counterparts, showed a very high degree of similarity in terms of genes expressed in the naive cells providing confidence that the iPSdMo, iPSdM and iPSdDC were suitable alternatives to PBMo, MDM and MoDC respectively. Such data has been generated previously for macrophages [28, 29] and the overall conclusion are similar to those presented in this study. While these studies also generated monocytes as precursors prior to generation of macrophages they did not perform transcriptomic analyses on this cell type.

Monocytes play an important role in disease biology and are targets of therapies in a number of diseases [39–42] and as such are an important cell type in their own right. While the

iPSdMo generated in this study show a high degree of similarity in gene expression compared to PBMo there were more genes that differed between the two than for either macrophages or DC. This may have been in part because these cells are still not as fully committed as either the macrophages or DC obtained by further differentiations treatment steps. Given the last 7 days of the differentiation protocol involves treatment with M-CSF (as is the case for differentiation of the macrophages) it is perhaps not surprising the MDM and iPSdM are most transcriptionally similar. Making changes to this final step of the monocyte differentiation may yield iPSdMo that more closely resemble the blood derived cells. Alternatively, further investigation of the CD14- cell population may identify a non-classical monocyte population.

Dendritic cells are key players in many diseases but there are known to be many different sub-classes of DC with different functional roles, both those from tissues as well as blood derived. We have been able to generate iPSdDC that share functional and transcriptional similarities with monocyte derived DC (MoDC) but a recent classification of blood DC subtypes using single cell sequencing [15] has allowed us to identify DC subtypes that are an even closer match transcriptionally to the iPSdDC. Of the 6 DC subtypes defined by Villani the blood derived DC (MoDC) are transcriptionally most similar to DC2, one of the groups of conventional DC (cDC) that express CD1C or DC1. The iPSdDC on the other hand appear to most closely resemble the DC5 or DC6/pDC subpopulation. The DC5 are a newly identified subset of blood DC present in small numbers that do not correspond to any of the previously known blood DC subtypes and are best defined by the surface markers AXL and SIGLEC6 [15]. This raises the possibility that the iPSdDC cells may provide a source of DC that are usually very difficult to acquire in large numbers. Further functional characterisation of these cells will be necessary to confirm this.

## Conclusions

We present a method that allows efficient and cheaper generation of large numbers of iPSC-derived macrophages and dendritic cells that are functionally and transcriptionally similar to blood derived cells. Genetic manipulation of the iPSC parental line before differentiation provides the opportunity to generate all three cell types- monocytes, macrophages and DC with specific genetic alterations providing cell tailored for specific assays and drug screens for which other sources of these cells are not suitable.

## Supporting information

**S1 File. S1–S8 Tables full lists of differentially expressed genes (S1-S5) and IPA results (S6-S8).** More details provided on sheet 1.
(XLSX)

**S1 Fig. Clustered heatmap of combined Villani DC single cell samples and DC samples from this study.** Genes are those from Villani study that discriminated the DC subgroups (S2 Table, AUC>0.85). Colours along X-axis define the sample groups (see key) and illustrate that the iPSdDC (cream) cluster between DC5 (dark blue) and DC6 (orange) while the MoDC (teal) cluster in DC1 (royal blue).
(TIF)

**S2 Fig. Venn diagrams illustrating the overlap in differentially expressed genes between the three pairs of comparison in this study.** A) Up regulated genes. Gene from each comparison that passed the cut off LFC>2, padj < 0.01. B) Down regulated genes. Gene from each comparison that passed the cut off LFC< -2, padj < 0.01.
(TIF)

## Acknowledgments

We would like to thank Graham Belfield for useful discussions on the design and execution of this work.

## Author Contributions

**Conceptualization:** Susan Monkley, Jayendra Kumar Krishnaswamy, Melker Göransson, Maryam Clausen, Johan Meuller, Kristofer Thörn, Ryan Hicks, Stephen Delaney, Louise Stjernborg.

**Data curation:** Susan Monkley.

**Formal analysis:** Susan Monkley, Jayendra Kumar Krishnaswamy, Melker Göransson, Johan Meuller, Ryan Hicks, Stephen Delaney, Louise Stjernborg.

**Investigation:** Susan Monkley, Jayendra Kumar Krishnaswamy, Melker Göransson, Johan Meuller, Kristofer Thörn, Louise Stjernborg.

**Methodology:** Susan Monkley, Jayendra Kumar Krishnaswamy, Maryam Clausen, Louise Stjernborg.

**Project administration:** Susan Monkley, Louise Stjernborg.

**Software:** Susan Monkley.

**Visualization:** Susan Monkley.

**Writing – original draft:** Susan Monkley, Jayendra Kumar Krishnaswamy, Stephen Delaney, Louise Stjernborg.

**Writing – review & editing:** Susan Monkley, Jayendra Kumar Krishnaswamy, Melker Göransson, Maryam Clausen, Johan Meuller, Kristofer Thörn, Stephen Delaney, Louise Stjernborg.

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
