## [Decision Letter · Decision Letter 0]

7 Aug 2020

PONE-D-20-21871

Optimised generation of iPSC-derived macrophages and dendritic cells that are functionally and transcriptionally similar to their primary counterparts

PLOS ONE

Dear Dr. Monkley,

Thank you for submitting your manuscript to PLOS ONE. After careful consideration, we feel that it has merit but does not fully meet PLOS ONE’s publication criteria as it currently stands. Therefore, we invite you to submit a revised version of the manuscript that addresses the points raised during the review process.

A rebuttal letter that responds to each point raised by the academic editor and reviewer(s). You should upload this letter as a separate file labeled 'Response to Reviewers'.A marked-up copy of your manuscript that highlights changes made to the original version. You should upload this as a separate file labeled 'Revised Manuscript with Track Changes'.An unmarked version of your revised paper without tracked changes. You should upload this as a separate file labeled 'Manuscript'.Please review the criteria for publication in PLOS One in regards to making transcriptomics data available. Raw gene expression data in this paper should be made available in a public repository to assure scientific and peer validation and reproducibility. Raw quantitative gene expression data should not include patient-specific identification, and cannot be identified in the manner patient-specific genomic information might, so consent and confidentiality should be protectable.

We look forward to receiving your revised manuscript.

Kind regards,

Elias T. Zambidis

Academic Editor

PLOS ONE

Journal Requirements:

2. We note that you are reporting an analysis of a microarray, next-generation sequencing, or deep sequencing data set. PLOS requires that authors comply with field-specific standards for preparation, recording, and deposition of data in repositories appropriate to their field. Please upload these data to a stable, public repository (such as ArrayExpress, Gene Expression Omnibus (GEO), DNA Data Bank of Japan (DDBJ), NCBI GenBank, NCBI Sequence Read Archive, or EMBL Nucleotide Sequence Database (ENA)). In your revised cover letter, please provide the relevant accession numbers that may be used to access these data. For a full list of recommended repositories, see http://journals.plos.org/plosone/s/data-availability#loc-omics or http://journals.plos.org/plosone/s/data-availability#loc-sequencing.

3. Thank you for stating the following in the Competing Interest section:

"AstraZeneca is a global biopharmaceutical company specialising in the discovery, development, manufacturing and marketing of prescription medicines. At the time of the study all authors were affiliated to AstraZeneca. "

We note that one or more of the authors are employed by commercial companies: 'AstraZeneca', 'Mölnlycke Healthcare' and 'Galderma Pharma SA'.

3.1. Please provide an amended Funding Statement declaring this commercial affiliation, as well as a statement regarding the Role of Funders in your study. If the funding organization did not play a role in the study design, data collection and analysis, decision to publish, or preparation of the manuscript and only provided financial support in the form of authors' salaries and/or research materials, please review your statements relating to the author contributions, and ensure you have specifically and accurately indicated the role(s) that these authors had in your study. You can update author roles in the Author Contributions section of the online submission form.

3.2. Please also provide an updated Competing Interests Statement declaring this commercial affiliation along with any other relevant declarations relating to employment, consultancy, patents, products in development, or marketed products, etc. 

Reviewers' comments:

Reviewer's Responses to Questions

**Comments to the Author**

1. Is the manuscript technically sound, and do the data support the conclusions?

Reviewer #1: Yes

Reviewer #2: Partly

Reviewer #3: Yes

2. Has the statistical analysis been performed appropriately and rigorously? 

Reviewer #1: Yes

Reviewer #2: No

Reviewer #3: N/A

3. Have the authors made all data underlying the findings in their manuscript fully available?

Reviewer #1: Yes

Reviewer #2: Yes

Reviewer #3: No

4. Is the manuscript presented in an intelligible fashion and written in standard English?

Reviewer #1: Yes

Reviewer #2: Yes

Reviewer #3: Yes

5. Review Comments to the Author

Reviewer #1: The authors have modified a previously described protocol for generating monocyte, macrophages, and dendritic cells from human iPSC under feeder-free conditions (and without serum until the monocyte stage) by initiating the cultures with embryoid bodies of specific size and concentration rather than use of monolayer cultures. Using this approach they achieve ~4-fold increased number of monocytes over the course of the culture that they collect from the supernatant. The authors further differentiate these to macrophages using M-CSF or dendritic cells using GM-CSF and IL-4 and then conduct functional analyses and triplicate RNAseq in comparison to monocytes from human peripheral blood and their maturation into macrophages or dendritic cells. iPSC-derived macrophages phagocytose S. aur and produce cytokines in response to LPS similar to blood-derived macrophages. iPSC-DCs were able to present ovalbumin, induce T cell proliferation (similar to blood-derived DCs), and release cytokines in response to LPS. RNAseq analysis showed marked similarity between iPSC and blood monocytes (85% of RNAs less than 45-fold different) and macrophages (95% similarity) and iPSC-DC and the DC5 and DC6 human DC subsets, with loss of pluripotency markers or markers of other blood lineages. Pathway analysis revealed some differences, e.g. with inflammatory response increased in iPSC monocytes and DCs and reduced in iPSC macrophages compared to blood-derived counterparts.

Overall, the authors provide a useful improvement in the production of monocytes, macrophages, and DCs from human iPSC and show that these cells fairly closely resemble blood-derived monocytes, macrophages, and DC5/6 cells. I have only minor comments:

1. Fig 2F legend says that cytokine product from DCs is shown for both iPSC-DC and blood-DCs, but only one data set is shown.

2. The blue shades for PBMo and MDM and the green shades for iPSdMo-CD14+ and iSPdM in Fig 4C are very similar.

Reviewer #2: 1.It is stated that the new protocol produces 2.5-4 x106 CD14+ monocytes per harvest and up to 3.5 x107 cells over the course of a single differentiation run which they claim compares very well to other recently published method. They say that the previous method used “multiple 100mm dishes” and the new method used a single 6 well plate. How many 100 mm dishes were used and do they move to a single well of a 6 well plate (ie one 35mm dish) or all 6 wells (6x3.5mm)? The authors should clarify and define the volume of media used to demonstrate their point.

It would be most appropriate to express the yield based on the starting number of iPSCs and discuss the actual reduction in media volume, thus addressing the cost of production more directly. If all this information is given then the reader would be more convinced that their new protocol is “higher yield and lower cost”.

2. The authors claim that CD14+ monocytic lineage-directed cells provide a continuous supply of cells that can be used directly or cryopreserved for later use but they do not show any evidence for their cryopreservation. Unless they can show convincing data for this point, they should remove this statement from the manuscript.

3. The differentiation run was performed three times to capture variability between differentiations and for each differentiation run 6-7 harvests of the suspension monocytes were collected (Fig. 1A). They claim that this allowed comparisons between runs and harvests to assess the reproducibility of the method. But the authors show no data for variability and reproducibility in the data presented in Figs 1 and 2? They should make it clear this point refers only to the transcriptomic data.

4. The authors claim that the phagocytic activity of iPSCdM is similar to MDM but the data indicate that there is a difference between the two cell types (Figure 2A). Over 80% MDMs phagocytosed but only 40-50% of iPSdM appeared to do so. The authors should plot the data for MDM and iPSdM on the same graph, perform statistical analysis to show how similar, or different, they are and then modify their conclusion accordingly. Other studies have shown that iPSCdM are less phagocytic than MDM. For example, Haidera et al (2017)(npj Regenerative Medicine 14) demonstrated that human iPSCdM phagocytosed less than MDM in both their naïve and polarized states. The result that the authors are showing here demonstrates a similar trend to that described by Haidera et al. The authors should acknowledge that, provide a possible explanation and cite the reference.

6. The authors make the statement,” like monocyte-derived dendritic cells, iPSdDC present antigen and release a range of cytokines following stimulation” . However, they do not show comparative data for monocyte-derived cells so it is not clear how they compare. The authors should show comparative data or modify their statement.

7. Figure 3A. The authors have been very selective in the genes that are shown in this heatmap. It is not surprising that genes associated with HSC, Granulocytes, B, T and NK cells are not expressed, given the fact that the differentiation protocol drives cells into the monocyte lineage. It would be far more interesting to include a larger number of genes associated with monocyte/macrophage and dendritic cells – the authors have cherry-picked only 2 marker genes for each which gives a misleading impression. Can this heatmap be extended to include more genes associated with these lineages? Also, the heatmap does not show the iPSdMo CD14- cell population for these markers?

8. Figure 4. The authors state that “only 565 genes were differentially expressed between iPSdM and MDM”. In my opinion this is quite a lot of genes and their differential expression could result in an altered phenotype. Again, the authors have cherry-picked the genes for the heatmap to create a specific impression. They could have equally chosen some of the differentially expressed genes to demonstrate the completely opposite result.

9. It is odd that genes activated in the iPSdM relative to MDM were mostly functions

associated with phagocytosis (Fig 5B) whereas the iPSdM actually appear LESS phagocytic (see my point about Figure 2A).

10. Overall, I think the authors are going to great lengths to emphasis similarities between iPSC-derived cells and peripheral blood monocyte-derived cells at the expense of considering the differences between these cell types. Although there are clearly a lot of expected similarities in gene expression profiles, it is perhaps the differences that are the more scientifically interesting points as addressing these differences will enable the scientific community to derived the most appropriate therapeutic cells types with optimal function. The authors make reference to this point in their discussion but the title of the manuscript and the statements in the abstract state they are similar. The paper would be more powerful if they state that they have done a comparison and found similarities AND interesting differences.

11. The authors should perform statistical analyses on data where conclusions have been suggested. This is important for the data shown in Figure 2A, B, D, E and F. They should also make a statement that the flow cytometry plots that are shown (eg Fig 1B and Fig 2C, D) are representative and provide some data on their reproducibility.

12. It is misleading to state that the method they have developed is cheap (Discussion). Although their modified protocol allows the production of cells in slightly lower volumes, it still depends on the use of costly media and cytokines at each stage over a long period of time. If they want to make a specific statement about relative cost, then they should show a breakdown of the costs and give an estimate about how much is saved using their protocol.

13. There is quite a bit of discussion in the literature that iPSC-derived macrophages are more similar to the macrophages of the primitive wave of hematopoiesis in the yolk sac that are thought to give rise to resident macrophages. The discussion should make reference to the developmental origin of tissue resident macrophages and speculate how this might impact on their data.

Minor points

1. interchangeable use of iPS-derived and iPSC-derived, the latter is correct

2. reference 30 refers to transcriptomics of beta cells, not blood-derived lineages as suggested.

Reviewer #3: The publication of Monkley et al describes an improved method for the generation of both macrophages and dendritic cells from human induced pluripotent stem cells. As mentioned by the authors, generation of macrophages and other blood cells is of high value not only for industry but also for regenerative and transfusion medicine in order to establish new therapies. Moreover, new and/or improved protocols would allow the investigation of developmental trajectories of human cells.

The manuscript is imbedded into a plethora of protocols generating macrophages and dendritic cells. Given the interesting data and especially the focus of the manuscript, it is surprising for me that the authors did not correlate their manuscript into the current landscape of other protocols dealing with macrophage/dendritic cell generation.

The manuscript is very well written and follows a logical path. The data analysis and presentation are very clear.

Please find below some comments, which may improve the overall quality of the manuscript.

Major:

1. The authors state an ethical study approval from the local ethical committee in Gothenburg. Given the affiliation of all authors with AstraZeneca the authors should provide information whether the ethical committee is also affiliated with AstraZeneca or whether it is belonging to the government, university, etc.

2. Figure 1B. The authors pre-gated their cells on CD14. In the CD14- fraction, why are there still CD14 SP and CD14/CD16 DP cells in the gating and what are CD14- cells?

3. I appreciate the transcriptome analysis performed in figure 3. The authors however should include non-hierarchical cluster analysis of their samples in order to get an unbiased overview of all samples. It is not surprising that all samples share no similarities with iPSC and thus all cells are very much different to them. Plotting a heatmap and non-hierarchical cluster analysis would provide a better overview than pre-selected genes.

4. Similarly, I would request a gene-ontology analysis especially for CD14+ to CD14- cells in order to get a better impression of the differences.

5. One of the main focus of the manuscript (also mentioned by the authors) is the improvement of an already published manuscript with respect to cell quantity/quality as well as economics. This being said, the discussion sounds very much as a summary including references to the figures. The authors should discuss their findings and correlate the findings with other protocols published. There is no link at all to other papers comparing the efficiency and/or quality. In addition, other publications have already compared both macrophages and DCs to primary counterparts, which have not been cited/used in the current manuscript. Given the focus of the manuscript, I highly recommend restructuring the discussion.

6. Although the authors declare the reason why not sharing their data, I am still wondering why this is the case. The manuscript does not show any genomics, which I could understand not sharing. Thus, the ethical/data sharing statement for me is confusing. This manuscript provides transcriptomic data, which is, at least to my understanding, impossible to trace back to an individual. Please also note, although the iPSC line was reprogrammed previously also by members of AstraZeneca, the original fibroblast for reprogramming have been obtained/purchased from ATCC. Thus, I would be surprised if AstraZeneca has a written inform consent from the donor not sharing this data (see statement “…Our first obligation is to

honour the contracts with our patients, only allowing access and use as agreed in the Informed Consent Forms...“ Whether or not AstraZeneca has the right to opt out of not sharing this data I cannot judge. However, given the strong focus of this manuscript on transcriptomics and the intention of the authors to publish their work in a scientific journal, I highly recommend uploading the data to a public database.

Minor

The legend of Figure 1 is in the materials and methods section

Reference 14 and 21 are the same. Please also check text.

Figure 1B: Did the authors also checked the more common myeloid marker CD11b on both CD14+/- fractions?

The decreased antigen presentation in figure 2D for iPSC-DC, is this also seen in monocyte -derived DCs, which in turn is in line with figure 2E?

Within the MLR and the data of figure 2F, would it be possible that the observation is rather an allo-reaction than a specific T cell activation. The authors should comment on this or provide a more detailed analysis of T cell activation.

Statistics, e.g. for figure 2 could be performed (e.g. 1-/2- way Anova?)

6. PLOS authors have the option to publish the peer review history of their article (what does this mean?). If published, this will include your full peer review and any attached files.

Reviewer #1: No

Reviewer #2: **Yes: **Lesley Forrester

Reviewer #3: No

---

## [Author Response · Author response to Decision Letter 0]

28 Oct 2020

Reviewer#1

Overall, the authors provide a useful improvement in the production of monocytes, macrophages, and DCs from human iPSC and show that these cells fairly closely resemble blood-derived monocytes, macrophages, and DC5/6 cells. I have only minor comments:

1. Fig 2F legend says that cytokine product from DCs is shown for both iPSC-DC and blood-DCs, but only one data set is shown.

Response: Text has been amended to reflect that data is from iPSdDC

2. The blue shades for PBMo and MDM and the green shades for iPSdMo-CD14+ and iSPdM in Fig 4C are very similar.

Response: Figure 4C has been altered accordingly. 

Reviewer #2

1. It is stated that the new protocol produces 2.5-4 x106 CD14+ monocytes per harvest and up to 3.5 x107 cells over the course of a single differentiation run which they claim compares very well to other recently published method. They say that the previous method used “multiple 100mm dishes” and the new method used a single 6 well plate. How many 100 mm dishes were used and do they move to a single well of a 6 well plate (ie one 35mm dish) or all 6 wells (6x3.5mm)? The authors should clarify and define the volume of media used to demonstrate their point.

It would be most appropriate to express the yield based on the starting number of iPSCs and discuss the actual reduction in media volume, thus addressing the cost of production more directly. If all this information is given then the reader would be more convinced that their new protocol is “higher yield and lower cost”.

Response: Additional text has been added to line 253 detailing yields in terms of specific volumes of media and culture surface area and how this compares to recent publications. It is not meaningful to express yields in terms of starting numbers of cells as the staring material is numbers of embryoid bodies (EB, for this method) and the numbers of cells is unknown. It is even more difficult to determine starting cell numbers for other methods as the EB are variable is size.

2. The authors claim that CD14+ monocytic lineage-directed cells provide a continuous supply of cells that can be used directly or cryopreserved for later use but they do not show any evidence for their cryopreservation. Unless they can show convincing data for this point, they should remove this statement from the manuscript.

Response: This statement has been removed from the manuscript. 

3. The differentiation run was performed three times to capture variability between differentiations and for each differentiation run 6-7 harvests of the suspension monocytes were collected (Fig. 1A). They claim that this allowed comparisons between runs and harvests to assess the reproducibility of the method. But the authors show no data for variability and reproducibility in the data presented in Figs 1 and 2? They should make it clear this point refers only to the transcriptomic data.

Response: 'Transcriptomic' has been introduced (~line 263). 

4. The authors claim that the phagocytic activity of iPSCdM is similar to MDM but the data indicate that there is a difference between the two cell types (Figure 2A). Over 80% MDMs phagocytosed but only 40-50% of iPSdM appeared to do so. The authors should plot the data for MDM and iPSdM on the same graph, perform statistical analysis to show how similar, or different, they are and then modify their conclusion accordingly. Other studies have shown that iPSCdM are less phagocytic than MDM. For example, Haidera et al (2017)(npj Regenerative Medicine 14) demonstrated that human iPSCdM phagocytosed less than MDM in both their naïve and polarized states. The result that the authors are showing here demonstrates a similar trend to that described by Haidera et al. The authors should acknowledge that, provide a possible explanation and cite the reference.

Response: The MDM and iPSdM data are now plotted on the same graph and include statistical comparison (Fig. 2A). The discussion has been adjusted accordingly including a reference to Haidera et al. 

6. The authors make the statement,” like monocyte-derived dendritic cells, iPSdDC present antigen and release a range of cytokines following stimulation” . However, they do not show comparative data for monocyte-derived cells so it is not clear how they compare. The authors should show comparative data or modify their statement.

Response: The Fig.2 legend has been altered to reflect which cells were measured and the text altered to reflect that the iPSdDC response is comparable to that in literature. 

7. Figure 3A. The authors have been very selective in the genes that are shown in this heatmap. It is not surprising that genes associated with HSC, Granulocytes, B, T and NK cells are not expressed, given the fact that the differentiation protocol drives cells into the monocyte lineage. It would be far more interesting to include a larger number of genes associated with monocyte/macrophage and dendritic cells – the authors have cherry-picked only 2 marker genes for each which gives a misleading impression. Can this heatmap be extended to include more genes associated with these lineages? Also, the heatmap does not show the iPSdMo CD14- cell population for these markers?

Response: The heatmap in Fig3A was generated to serve a particular purpose, to demonstrate that the differentiation protocol had achieved the desired outcome in terms of generating specifically monocyte derived cells as determined by the gene expression markers (the text has been reworded ~line 341 to reflect this). The genes chosen for the heatmap were those commonly used to identify/select for the cell types listed and are commonly used to do so. This figure was not intended as a in depth exploration of the expression profiles of the iPSC derived cell types. This is addressed in the rest of the manuscript and subsequent figures. Upon establishing the differentiation method worked (something we felt is important before continuing the analysis) we investigated all the genes via the global RNAseq analysis and differential expression and we provide full gene lists of expression and expression changes (Supplementary and deposited data). This allows readers to explore the genes of their choice rather than restricting the list to the few marker genes that can be fitted on a heatmap. The heatmap does no include the iPSdMo_CD14- as this cell type is not one we further investigated for reasons explained in answer to comments 3.2 and 3.4

8. Figure 4. The authors state that “only 565 genes were differentially expressed between iPSdM and MDM”. In my opinion this is quite a lot of genes and their differential expression could result in an altered phenotype. Again, the authors have cherry-picked the genes for the heatmap to create a specific impression. They could have equally chosen some of the differentially expressed genes to demonstrate the completely opposite result.

Response: The reason for the use of the phrase “only 565 genes were differentially expressed between iPSdM and MDM” is that of all 3 comparisons made in this article (iPSdMo vs PBMo, iPSdM vs MDM an iPSdDC vs MoDC) the number of differentially expressed genes was lowest between iPSdM and MDM. In addition while 565 genes were differentially expressed, over 10000 genes were not. Despite that we do fully accept that 565 altered genes could and do lead to an altered phenotype which we explore in Fig 5B with the functional enrichment analysis highlighting what the differences may lead to in terms of functional changes in the iPSdM compared to MDM such as phagocytosis which we discuss in detail in the discussion.

9. It is odd that genes activated in the iPSdM relative to MDM were mostly functions associated with phagocytosis (Fig 5B) whereas the iPSdM actually appear LESS phagocytic (see my point about Figure 2A).

Response: Of the 3 functions associated with genes activated in the iPSdM relative to MDM, 2 could be associated with phagocytosis (engulfment of cells and endocytosis) however this does not necessarily mean the genes in these 2 functional groups are positively associated with these functions. It is just as possible they are negative regulators of the processes.

10. Overall, I think the authors are going to great lengths to emphasis similarities between iPSC-derived cells and peripheral blood monocyte-derived cells at the expense of considering the differences between these cell types. Although there are clearly a lot of expected similarities in gene expression profiles, it is perhaps the differences that are the more scientifically interesting points as addressing these differences will enable the scientific community to derived the most appropriate therapeutic cells types with optimal function. The authors make reference to this point in their discussion but the title of the manuscript and the statements in the abstract state they are similar. The paper would be more powerful if they state that they have done a comparison and found similarities AND interesting differences.

Response: It is not our intention to overstate the similarities as we do recognize and expect there to be differences. Our intention is to provide a balanced overview and then also make the data available to allow others to decide for themselves if the similarities and or differences are most relevant depending in the functional requirement of the cells. We have therefore adjusted the wording to be more balanced. We expanded on the reasons for differences some of which are potentially of great interest eg rare DC subtypes. We have highlighted these differences in moncytes and DCs as well. 

11. The authors should perform statistical analyses on data where conclusions have been suggested. This is important for the data shown in Figure 2A, B, D, E and F. They should also make a statement that the flow cytometry plots that are shown (eg Fig 1B and Fig 2C, D) are representative and provide some data on their reproducibility.

Response: Statistical tests have been added to plots in Fig. 2. 

12. It is misleading to state that the method they have developed is cheap (Discussion). Although their modified protocol allows the production of cells in slightly lower volumes, it still depends on the use of costly media and cytokines at each stage over a long period of time. If they want to make a specific statement about relative cost, then they should show a breakdown of the costs and give an estimate about how much is saved using their protocol.

Response: Specific savings have been presented in the Results and wording in the Discussion and Conclusion has been changed from cheap to cheaper. 

13. There is quite a bit of discussion in the literature that iPSC-derived macrophages are more similar to the macrophages of the primitive wave of hematopoiesis in the yolk sac that are thought to give rise to resident macrophages. The discussion should make reference to the developmental origin of tissue resident macrophages and speculate how this might impact on their data.

Response: Additions to address this point have been added to the Discussion starting on line 466. 

Minor points

1. interchangeable use of iPS-derived and iPSC-derived, the latter is correct

2. reference 30 refers to transcriptomics of beta cells, not blood-derived lineages as suggested.

Response: These inconsistencies have been corrected. 

Reviewer #3: 

Major:

1. The authors state an ethical study approval from the local ethical committee in Gothenburg. Given the affiliation of all authors with AstraZeneca the authors should provide information whether the ethical committee is also affiliated with AstraZeneca or whether it is belonging to the government, university, etc.

Response: Details of the Local Ethical committee have been added to Methods. 

2. Figure 1B. The authors pre-gated their cells on CD14. In the CD14- fraction, why are there still CD14 SP and CD14/CD16 DP cells in the gating and what are CD14- cells?

Response: The cells were not pre-gated as such as would be the case in FACS sorted cells. The sorting step for the CD14+ cells that occurs prior to the differentiation into macrophages or DCs is a magnetic activated cell sorting (MACS) rather than FACS (fluorescent activated cell sorting). With MACS the CD14 antibody is bound to magnetic beads and these are used to capture and purify the CD14 expressing cells. However due to the nature of the procedure there will be cells that express CD14 especially at lower levels that do not bind sufficiently strongly to the magnetic beads to be retained and will end up in the CD14- population after this step. We did not investigate the nature of the CD14- cells any further than this as is discussed in comment 3.2 as it was outside the scope if this study. 

3. I appreciate the transcriptome analysis performed in figure 3. The authors however should include non-hierarchical cluster analysis of their samples in order to get an unbiased overview of all samples. It is not surprising that all samples share no similarities with iPSC and thus all cells are very much different to them. Plotting a heatmap and non-hierarchical cluster analysis would provide a better overview than pre-selected genes.

Response: It is not clear why non-hierarchical cluster analysis of the samples would be a better method that the PCA (principal component analysis) in Fig 3B to get an overview of all samples and their relationships to one another. PCA plots are a much more common way of understanding the sample relationships associated with RNA sequencing data than non-hierarchical cluster methods and is hence the reason for our choice. We have worded the text (line 347) to make this clearer. 

4. Similarly, I would request a gene-ontology analysis especially for CD14+ to CD14- cells in order to get a better impression of the differences.

Response: We do not consider the CD14- population of sufficient biological interest to warrant this. As discussed in the text (line 470) the CD14- population is likely a mixed population of undifferentiated cells of limited biological relevance except perhaps in relation to what it could suggest about how the method could be improved to increase yield.

5. One of the main focus of the manuscript (also mentioned by the authors) is the improvement of an already published manuscript with respect to cell quantity/quality as well as economics. This being said, the discussion sounds very much as a summary including references to the figures. The authors should discuss their findings and correlate the findings with other protocols published. There is no link at all to other papers comparing the efficiency and/or quality. In addition, other publications have already compared both macrophages and DCs to primary counterparts, which have not been cited/used in the current manuscript. Given the focus of the manuscript, I highly recommend restructuring the discussion.

Response: Paragraph 1 Discussion has been amended in line with this and additional references have been added. This is also addressed in the Results in more detail referencing recent new methods (see also response to Reviewer#2 Points 1, 12 and 13). 

6. Although the authors declare the reason why not sharing their data, I am still wondering why this is the case. The manuscript does not show any genomics, which I could understand not sharing. Thus, the ethical/data sharing statement for me is confusing. This manuscript provides transcriptomic data, which is, at least to my understanding, impossible to trace back to an individual. Please also note, although the iPSC line was reprogrammed previously also by members of AstraZeneca, the original fibroblast for reprogramming have been obtained/purchased from ATCC. Thus, I would be surprised if AstraZeneca has a written inform consent from the donor not sharing this data (see statement “…Our first obligation is to

honour the contracts with our patients, only allowing access and use as agreed in the Informed Consent Forms...“ Whether or not AstraZeneca has the right to opt out of not sharing this data I cannot judge. However, given the strong focus of this manuscript on transcriptomics and the intention of the authors to publish their work in a scientific journal, I highly recommend uploading the data to a public database.

Response: The data has been deposited in ArrayExpress accession number E-MTAB-9670 with the raw (fastq) files deposited for the cell line samples and the processed data (counts/TPM) deposited for all samples. 

Minor

The legend of Figure 1 is in the materials and methods section

Response: This is in keeping with the PLOS ONE requirement that the Figure (and therefore legend) appear after first reference in the text. The first reference to Fig. 1 is in the Methods.

Reference 14 and 21 are the same. Please also check text.

Response: Corrected. 

Figure 1B: Did the authors also checked the more common myeloid marker CD11b on both CD14+/- fractions?

Response: No we do not routinely use this antibody.

The decreased antigen presentation in figure 2D for iPSC-DC, is this also seen in monocyte -derived DCs, which in turn is in line with figure 2E?

Response: The reduction in antigen uptake in presence of LPS we discuss may be due to differences in phagosome maturation in TLR activated iPSdDC (line 488) but this does not alter the antigen presentation capacity as assessed by the MLR assay (Fig. 2E). 

Within the MLR and the data of figure 2F, would it be possible that the observation is rather an allo-reaction than a specific T cell activation. The authors should comment on this or provide a more detailed analysis of T cell activation.

Response: The assay used by us to evaluate antigen presentation is a Mixed Lymphocyte reaction and has conventionally been used to evaluate antigen presentation by human DC populations (now referenced in manuscript text). While the response is not antigen specific, the assay uses purified T cells and DCs (IPS derived or monocyte-derived DCs) and is mediated by MHCII-TCR interactions. (see also response to Reviewer#2 Point 6). 

Statistics, e.g. for figure 2 could be performed (e.g. 1-/2- way Anova?)

Response: These have been added to Figure 2.

---

## [Decision Letter · Decision Letter 1]

27 Nov 2020

Optimised generation of iPSC-derived macrophages and dendritic cells that are functionally and transcriptionally similar to their primary counterparts

PONE-D-20-21871R1

Dear Dr. Monkley,

We’re pleased to inform you that your manuscript has been judged scientifically suitable for publication and will be formally accepted for publication once it meets all outstanding technical requirements.

Kind regards,

Elias T. Zambidis

Academic Editor

PLOS ONE

Additional Editor Comments (optional):

Reviewers' comments:

Reviewer's Responses to Questions

**Comments to the Author**

1. If the authors have adequately addressed your comments raised in a previous round of review and you feel that this manuscript is now acceptable for publication, you may indicate that here to bypass the “Comments to the Author” section, enter your conflict of interest statement in the “Confidential to Editor” section, and submit your "Accept" recommendation.

Reviewer #1: All comments have been addressed

Reviewer #2: All comments have been addressed

Reviewer #3: (No Response)

2. Is the manuscript technically sound, and do the data support the conclusions?

Reviewer #1: Yes

Reviewer #2: Yes

Reviewer #3: Partly

3. Has the statistical analysis been performed appropriately and rigorously? 

Reviewer #1: Yes

Reviewer #2: Yes

Reviewer #3: Yes

4. Have the authors made all data underlying the findings in their manuscript fully available?

Reviewer #1: Yes

Reviewer #2: Yes

Reviewer #3: Yes

5. Is the manuscript presented in an intelligible fashion and written in standard English?

Reviewer #1: Yes

Reviewer #2: Yes

Reviewer #3: Yes

6. Review Comments to the Author

Reviewer #1: (No Response)

Reviewer #2: The authors have provided a comprehensive response to my comments and have submitted a much improved version of the manuscript. They have altered the manuscript to answer my concerns and/or have reworded statements to better reflect the data they present. This is a very nice paper that will be really useful for researchers in the field.

Reviewer #3: I am still not in line with the authors on presenting selected transcriptome datasets. My concerns/suggestions are in line with R#2, which have not been changed as well.

Furthermore, the authors still work in a allogenic setting and cannot rule out any allo-reactions in their assays.

For both points raised, the authors argued/replied in the point-by-point response, which would be fine for me to accept.

7. PLOS authors have the option to publish the peer review history of their article (what does this mean?). If published, this will include your full peer review and any attached files.

Reviewer #1: No

Reviewer #2: **Yes: **Lesley Forrester

Reviewer #3: No

---

## [Editor Report · Acceptance letter]

4 Dec 2020

PONE-D-20-21871R1 

Optimised generation of iPSC-derived macrophages and dendritic cells that are functionally and transcriptionally similar to their primary counterparts 

Dear Dr. Monkley:

I'm pleased to inform you that your manuscript has been deemed suitable for publication in PLOS ONE. Congratulations! Your manuscript is now with our production department. 

Kind regards, 

on behalf of

Dr. Elias T. Zambidis 

Academic Editor

PLOS ONE